# Evaluation of SARS-CoV-2 ORF7a Deletions from COVID-19-Positive Individuals and Its Impact on Virus Spread in Cell Culture

**DOI:** 10.3390/v15030801

**Published:** 2023-03-21

**Authors:** Maria Clara da Costa Simas, Sara Mesquita Costa, Priscila da Silva Figueiredo Celestino Gomes, Nádia Vaez Gonçalves da Cruz, Isadora Alonso Corrêa, Marcos Romário Matos de Souza, Marcos Dornelas-Ribeiro, Tatiana Lucia Santos Nogueira, Caleb Guedes Miranda dos Santos, Luísa Hoffmann, Amilcar Tanuri, Rodrigo Soares de Moura-Neto, Clarissa R. Damaso, Luciana Jesus da Costa, Rosane Silva

**Affiliations:** 1Instituto de Biofísica Carlos Chagas Filho, Universidade Federal do Rio de Janeiro, Rio de Janeiro 21941-902, Brazil; 2Instituto de Microbiologia Paulo de Góes, Universidade Federal do Rio de Janeiro, Rio de Janeiro 21941-902, Brazil; 3Physics Department, Auburn University, Auburn, AL 36849, USA; 4Laboratório de Biodefesa, Instituto de Biologia do Exército, Rio de Janeiro 20911-270, Brazil; 5Instituto Federal de Educação, Ciência e Tecnologia do Rio de Janeiro, Rio de Janeiro 20270-021, Brazil; 6Instituto de Biologia, Universidade Federal do Rio de Janeiro, Rio de Janeiro 21941-902, Brazil

**Keywords:** SARS-CoV-2, ORF7a, COVID-19, sgRNAs, viral fitness, B.1.1.33 lineage

## Abstract

The spread of severe acute respiratory syndrome coronavirus 2 (SARS-CoV-2), causing the COVID-19 outbreak, posed a primary concern of public health worldwide. The most common changes in SARS-CoV-2 are single nucleotide substitutions, also reported insertions and deletions. This work investigates the presence of SARS-CoV-2 ORF7a deletions identified in COVID-19-positive individuals. Sequencing of SARS-CoV-2 complete genomes showed three different ORF7a size deletions (190-nt, 339-nt and 365-nt). Deletions were confirmed through Sanger sequencing. The ORF7a∆190 was detected in a group of five relatives with mild symptoms of COVID-19, and the ORF7a∆339 and ORF7a∆365 in a couple of co-workers. These deletions did not affect subgenomic RNAs (sgRNA) production downstream of ORF7a. Still, fragments associated with sgRNA of genes upstream of ORF7a showed a decrease in size when corresponding to samples with deletions. In silico analysis suggests that the deletions impair protein proper function; however, isolated viruses with partial deletion of ORF7a can replicate in culture cells similarly to wild-type viruses at 24 hpi, but with less infectious particles after 48 hpi. These findings on deleted ORF7a accessory protein gene, contribute to understanding SARS-CoV-2 phenotypes such as replication, immune evasion and evolutionary fitness as well insights into the role of SARS-CoV-2_ORF7a in the mechanism of virus-host interactions.

## 1. Introduction

The COVID-19 outbreak, caused by the severe acute respiratory syndrome coronavirus 2 (SARS-CoV-2), is a major concern of public health that is rapidly spreading worldwide. So far, the virus has infected over 646 million people in Brazil, and over 6.6 million deaths have been reported globally (https://www.who.int/emergencies/diseases/novel-coronavirus-2019/situation-reports [accessed on 14 December 2022]). Most coronaviruses that are pathogenic to humans are associated with mild clinical symptoms [1,2], except for SARS-CoV-2, SARS-CoV and MERS-CoV. The previous SARS-CoV emerged in China in November 2002, causing more than 8000 human infections and 774 deaths in 37 countries; and the Middle East respiratory syndrome coronavirus (MERS-CoV) [3], detected in Saudi Arabia in 2012 [4], was responsible for more than 2000 cases of infection and 858 deaths. SARS-CoV-2 belongs to the Coronaviridae family and is similar to other coronaviruses [5,6]. The SARS-CoV-2 genome is a positive single-stranded RNA of nearly 30,000 nucleotides. At the 5′ end, the genomic RNA (gRNA) features two large open reading frames (ORFs; ORF1a and ORF1b) that encode 16 non-structural proteins (nsp), among them the viral replication and transcription complex [7]. ORFs that encode structural and accessory proteins are transcribed from the 3′ one-third of the genome, forming a nested set of subgenomic RNAs (sgRNA), which are responsible for encoding Spike (S), Envelope (E), Membrane (M), Nucleocapsid (N) and 3a, 6, 7a, 7b, 8 and 10 accessory proteins [8,9].

SgRNAs are synthesized during the production of the negative strand, in which the RNA-dependent RNA polymerase switches template to Transcription Regulatory Sequences (TRS) located upstream of the ORFs on the third, final part of the virus genome. Because of the complementarity of the TRS-Body (TRS-B) with TRS-Leader (TRS-L), the polymerase suffers a template switch to the TRS-L that stands on the 5′ of the genomes [10]. The resulting negative RNA strand contains a fused sequence of TRS-L and TRS-B, which will serve as a template for mRNA transcription [9]. Since sgRNAs are only transcribed in infected cells, they serve as indicators of active replication [11] and their reduced expression has been associated with mild COVID-19 cases [12]. The most common changes in SARS-CoV-2 are single nucleotide substitutions (SNSs), although insertions and deletions have also been reported [13,14]. Single nucleotide variants (SNVs) are found in various sites on non-structural genes that code for 16 nsp on the first 2/3 of the SARS genome as well on structural proteins as S, which is responsible for recognition of the host receptor protein [15,16]. Most of these single nucleotide variants cause synonymous mutation [17], which may be selected due to a better fit or increasing pathogenicity. A mutation rate of 1.12 × 10^−3^ nucleotides/per site/per year [17] was calculated in SARS-CoV-2, SARS-CoV and MERS. SNVs in the S gene have been associated with the genomic evolution of the virus, allowing tracking of emerging variants with potential impact on virus control and vaccine development [18]. Indels are also interesting to study because they may represent evidence of ongoing virus adaptation to human hosts through natural attenuation [19,20].

SARS-CoV-2_ORF7a, the ortholog of SARS-CoV_ORF7a, is encoded by the 366 nucleotides long orf7a gene, a type I transmembrane protein, and plays an essential role in virus–host interactions. Data suggest that SARS-CoV Ig-like viral protein ORF7a interacts with immune cells such as the integrin LFA-1 and CD14+ monocytes [21,22,23,24], acting as an immunomodulating factor for immune cell binding and triggering dramatic inflammatory responses.

Studies on SARS-CoV-2 deleted from ORF7a have contributed to understanding SARS-CoV-2 replication and shed light on the mechanisms of virus–host interactions. A few deletions have been reported for ORF7a and adjacencies [25,26,27,28,29,30] varying in size from 4 to 454 nucleotides long, but the impact of such deletions was not investigated. This study aimed to investigate the presence of ORF7a deletions in SARS-CoV-2 isolated from individuals and to explore their impact on virus replication in cell culture. From April to September 2020, we encountered three different ORF7a size deletions (190-nt, 339-nt and 365-nt) in the Rio de Janeiro samples analyzed. The ORF7a_∆190 was detected in a group of five relatives with mild cases of COVID-19, and the ORF7a_∆339 and ORF7a_∆365 in a couple of co-workers. ORF7a_∆190 deletion severely impacts the protein structure and function by depleting a significant portion of ORF7a ectodomain, transmembrane domain and ER retention motif. Viruses with ORF7a deletions showed reduced production of infectious particles in culture cells after 48 and 72 h of infection.

## 2. Materials and Methods

### 2.1. Samples

Biological samples were obtained using three synthetic nasal and oropharyngeal swabs (nose—2, throat—1) collected from May to June 2020 at the Army Biology Institute (IBEX), Rio de Janeiro, Brazil. Samples from the Diagnostic Center of UFRJ (CTD-UFRJ) were collected from August to September 2020. Sample selection criteria for deep sequencing were high viral load, different age groups, different locations and individuals exerting the same work activity. Approval from the Research Ethics Committee of Centro de Capacitação Física do Exército (CCFEx) (30918520.4.0000.9433) and local ethics review committee from Clementino Fraga Filho University Hospital (CAAE 30161620.0.0000.5257).

### 2.2. Cell Culture

African green monkey kidney cells (Vero E6; ATCC CRL-1586) and African green monkey kidney cells expressing human Transmembrane Serine Protease 2 and human Angiotensin-Converting Enzyme 2 (Vero E6-TMPRSS2-T2A-ACE2 cells; NR-54970) were maintained in DMEM with 4.5 g/L of D-Glucose (Dulbecco’s Modified Eagle’s Medium—Gibco^TM^, Billings, MT, USA), 1 mM sodium pyruvate, supplemented with 10% fetal bovine serum (ThermoFisher Scientific, Waltham, MA, USA), and 0.75% *w/v* sodium bicarbonate. Human lung adenocarcinoma epithelial cells (Calu-3, ATCC HTB-55) were maintained in DMEM with 1.0 g/L of D-Glucose (Dulbecco’s Modified Eagle’s Medium—Gibco^TM^, Billings, MT, USA), 1mM sodium pyruvate, supplemented with 10% fetal bovine serum (ThermoFisher Scientific, Waltham, MA, USA), and 0.75% *w/v* sodium bicarbonate. Cells were cultured at 37 °C with 5% CO_2_.

### 2.3. SARS-CoV-2 Isolation from Clinical Samples

Nasopharyngeal or oropharyngeal swab samples collected from positive patients for SARS-CoV-2 were added in 1 mL DMEM, followed by filtration in a 0.22 μm PES filter. A volume of 0.5 mL of the filtered virus was mixed with 2.5 mL of non-supplemented DMEM and added to a monolayer of Vero E6 cells, kept in a 75 cm^2^ bottle. The cells were incubated for 1 h at 37 °C, in an atmosphere with 5% CO_2_ for viral adsorption. After 1 h of incubation, the medium was changed by supplemented DMEM, and the cells were incubated at 37 °C, in an atmosphere with 5% CO_2_, for 72 h or until visualization of the cytopathic effect. Cell supernatant was collected, filtered in a 0.22 μm PES filter, and fractionated in microtubes. Tubes containing the isolated virus were kept at −80 °C.

### 2.4. Vero E6 Infection

Vero E6 cells were plated in 24-well to achieve 90% confluence overnight. Cells were infected with SARS-CoV-2_ORF7a_wt and SARS-CoV-2_ORF7a_Δ190 with a multiplicity of infection of 0.1 (MOI 0.1). After one hour of incubation, at 37 °C for virus adsorption, the cell medium was replaced with 10% FBS-supplemented DMEM. Cells were incubated at 37 °C, and 5% CO_2_, for up to 72 h. Cells supernatant were collected 24, 48 and 72 h post-infection (hpi), for virus titration and RT-qPCR. Molecular quantification was performed using the Detection Kit for 2019 Novel Coronavirus (2019-nCoV) RNA (Da An Gene Co., Ltd. of Sun Yat-sen University), according to the manufacturer’s instructions.

### 2.5. Viral Titration

Serial dilution of the viral stock was performed (10-1 to 10-6). A volume of 0.2 mL of the viral stock dilutions was applied to a monolayer of Vero E6 cells maintained in a 12-well plate. The cells were incubated for 1 h, at 37 °C, in an atmosphere with 5% CO_2_. After that incubation period, the virus’s medium was removed, and 1.0 mL of DMEM containing 1.4% carboxymethylcellulose (CMC) and 1% fetal bovine serum (FBS) was added. The cells were incubated at 37 °C, in an atmosphere with 5% CO^2^ for 72 h. Then, 1.0 mL of 10% formaldehyde was applied per well, and cells were incubated at room temperature for at least 3 h. The wells containing the cells were washed with water until the complete removal of the medium. Then 0.5 mL of 1% crystal violet, diluted in 20% methanol, was added. Cells were incubated for 5 to 10 min, at room temperature, followed by washing with water until the removal of the crystal violet excess. The formed plaques were counted, and the calculation was performed to determine the viral titer.

### 2.6. Statistical Analysis

For each of the time points post-infection (24, 48 and 72 hpi), titers of three replicates of SARS-CoV-2_wt and SARS-CoV-2_ORF7a_Δ190 expressed as SARS-CoV-2 PFU/mL and SARS-CoV-2 RNA equivalent to PFU were compared using analysis of multiple unpaired t-tests in GraphPad Version 9.1.0 and 8.0.2.263 (GraphPad Software, San Diego, CA, USA).

### 2.7. RNA Extraction and Viral Genome Quantification

RNAs were extracted using QIAmp RNA Viral mini kit and QIAcube automated platform (Qiagen, Hilden, Germany). Total RNA was quantified using Qubit 2.0 Fluorometer (Invitrogen, Waltham, MA, USA) with Qubit™ RNA HS Assay Kit (Invitrogen, Waltham, MA, USA).

### 2.8. cDNA Synthesis

cDNA synthesis was performed on 10 ng of RNA extracted from the positive samples. RNA was incubated at 70 °C, for 5 min, and then rapidly cooled on ice. Reverse transcription reaction was performed using SuperScript™ VILO™ cDNA Synthesis Kit (Thermofisher Scientific, Carlsbad, CA, USA), according to the manufacturer’s instructions.

### 2.9. ORF7a Targeted Amplification

A polymerase chain reaction of viral cDNA was carried out using Platinum™ Taq DNA Polymerase (Invitrogen) or GoTaq^®^ DNA Polymerase (Promega, Madison, WI, USA) following the manufacturer’s instructions. An amount of 1.25 U of enzyme and 2.5 µL of cDNA was used in each reaction. PCR product was obtained using the set of primers M_27145 F (5′-CAGACCATTCCAGTAGCA-3′ as the forward) and ORF8_28003 R (5′-CACGGGTCATCAACTACA-3′ as the reverse) at a final concentration of 0.2 µM that targeted the ORF7a region.

### 2.10. Sanger Sequencing of ORF7a Target Region

Amplification products were sequenced using BigDye™ Terminator v3.1 Cycle Sequencing Kit (Applied Biosystems, Waltham, MA, USA) following the manufacturer’s protocol and run into the 3500 ABI Genetic analyzer (Thermofisher Scientific, Carlsbad, CA, USA). Primers used in Sanger sequencing were the same used for targeted amplification but diluted accordingly.

### 2.11. sgRNA Targeted Amplification

cDNA from clinical samples was amplified with primers designed to produce a range of sgRNAs: the forward primer located on TRS leader region (SARS-Leader 18-41_F 5′-TCCCAGGTAACAAACCAACCAACT-3′), and the Reverse primer on the ORF8 or N regions (ORF8_27986-28003_R 5′-CACGGGTCATCAACTACA-3′; N_28529-28508_R 5′-AGCCAATTTGGTCATCTGGACT-3′). For the amplification reaction, a final concentration of 0.2 µM of each primer was used, 2.5 µl of cDNA was added, and GoTaq^®^ Long PCR Master Mix (Promega, USA) at a final concentration of 1X. Thermal cycling conditions were: polymerase activation at 94 °C for 2 min, followed by 35 cycles of denaturation at 94 °C for 20 s, annealing at 55 °C for 30 s, extension at 72 °C for 7 min, and a final extension step at 72 °C for 10 min. Amplicons were visualized through 2% agarose gel and stained with ethidium bromide. PCR fragments were extracted using E-Gel™ SizeSelect™ II Agarose Gel, 2% (Thermofisher Scientific, Carlsbad, CA, USA) and submitted to Sanger sequencing. The analysis and image of the sequences were generated in Geneious Prime 2022.2.2 (www.geneious.com [accessed on 31 December 2022]).

### 2.12. Next-Generation Sequencing (NGS)

NGS was achieved using the Ion AmpliSeq SARS-CoV-2 Research Panel (Thermofisher Scientific, Carlsbad, CA, USA), which provides complete coverage of the SARS-CoV-2 genome and variants, possible through the two sets of primer pools that cover the virus genome. The multiplex amplification reaction was performed using 10 µL of the cDNA according to the manufacturer’s instructions for 21 cycles of the multiplex RT-PCR-specific SARS-CoV-2 primers from the panel. Quantification of the libraries of each sample was obtained using Ion Library TaqMan™ Quantitation Kit (Thermofisher Scientific, Carlsbad, CA, USA). The pool of libraries at a concentration of 50pM was run on the Ion Chef™ system (ThermoFisher Scientific, Carlsbad, CA, USA) using Ion 510, 520 and 530 Kits (ThermoFisher Scientific, Carlsbad, CA, USA) and run into a 530 chip in an Ion S5™ System genetic sequencer (Thermofisher Scientific, Carlsbad, CA, USA).

### 2.13. Genome Analysis and Comparison

Reads generated were mapped to a SARS-CoV-2 reference genome Wuhan (NCBI GenBank accession number MN908947) using the Ion Browser software included in the Torrent Suite 5.10.1, and visualized in Integrative Genomic Viewer (IGV 2.6.3) (Broad Institute, Cambridge, MA, USA) [31]. Lineage and genome variants were accessed using the web tool Nextclade (c) 2020–2022 version 1.14.0 (Nextstrain developers) [32] through fasta files generated from NGS sequencing.

A phylogenetic analysis was performed using the sequenced SARS-CoV-2 samples and previously published 208 complete genome sequences, high coverage, available on GISAID, from April to September 2020, in Rio de Janeiro. Sequences were aligned using MAFFT [33] and submitted to IQ-TREE2 for maximum likelihood (ML) method of analysis [34]. The substitution model was General Time Reverse, and a proportion of invariable sites was selected (GTR + I), bootstrap of 1000 replicas, and tree method as Neighbor-Joining. The tree was visualized using CLC genomics workbench V.22.02 (digitalinsights.qiagen.com [accessed on 31 December 2022]).

### 2.14. Protein Sequence and Structural Analysis

The reference protein sequence for SARS-CoV-2 ORF7a was retrieved from the NCBI (accession number: YP009724395.1). Our sample containing the ORF7a deletion, here identified as 58301_ORF7a_∆190, was translated using Geneious Prime 2022.2.2 (www.geneious.com [accessed on 31 December 2022]). The protein sequences corresponding to the crystal structures of SARS-CoV-2 ORF7a (PDB ID: 7ci3) and SARS-CoV-1 ORF7a (PDB ID: 1xak) were retrieved from the Protein Data Bank (PDB) [35]. The protein sequences were aligned to the reference sequence using MAFFT [36] followed by manual adjustment of gaps. The AI-based protein predictor AlphaFold2 [37] was used to model the full-length structure of SARS-CoV-2 ORF7a. The best-ranked model was selected, and the tridimensional structure was used to map the deletions using PyMOL [38].

### 2.15. STR-Typing Profile

Nucleic acid extracted from the swab sample was quantified using the Qubit dsDNA broad-range quantification kit (Invitrogen, Madison, WI, USA). PCR products were run in a 3500 ABI genetic analyzer (Applied Biosystems, Carlsbad, CA, USA). STR profile was performed with Power Plex Fusion 6C (Promega, Madison, WI, USA), using 0.1 ng of DNA according to protocol. Profile was determined with the software GeneMapper IDX v.1.4.

## 3. Results

### 3.1. Identification of SARS-CoV-2 ORF7a Deletion in Family Members

RNA extracted from 36 nasopharyngeal or oropharyngeal swabs of outpatients collected by IBEX, who tested positive for SARS-CoV-2 were submitted to whole genome sequencing as a part of a collaboration for COVID-19 surveillance in Rio de Janeiro, Brazil. Most of the samples belonged to SARS-CoV-2 lineage B1.1.33. In one of the samples, we found a 190-nucleotide deletion in the ORF7a gene. To evaluate the deletion impact on viral spread, other family members who have tested positive for COVID-19 were contacted. Appendix A shows the clinical information all family members who had mild symptoms and without the need for hospitalization. Swab samples were collected and tested for ORF7a deletion through amplification using specific primers targeting the ORF7a gene. The forward primer is in the gene that encodes the M protein, and the reverse primer is in the gene corresponding to the accessory protein ORF8 (Figure 1A). Amplicons were generated and analyzed on agarose gel (Figure 1A). Samples were compared to an ORF7a wild-type SARS-CoV-2 B1.1.33 isolated from a human swab (sample here identified as 11785_ORF7a_wt—GISAD accession ID EPI_ISL_492038). We observed that all samples from family members had the same 190-nt deletions (ORF7a_∆190). We Sanger-sequenced the amplicons to confirm the deletions and access of the exact extent of deletion (position 27,512 to 27,701), as shown in Figure 1B.

We investigated the impact of the ORF7a deletion on the ORF7a protein structure. Similar to SARS-CoV_ORF7a, SARS-CoV-2_ORF7a is a type-I transmembrane protein with 121 amino acid residues, consisting of an N-terminal signaling region (residues 1–15), an Ig-like ectodomain (residues 16–96), a hydrophobic transmembrane domain (residues 97–116), and a typical ER retention motif (residues 117–121) (Figure 1C). Multiple sequence alignment of 58301_ORF7a_∆190 with ORF7a reference sequence and SARS-CoV and SARS-CoV-2 ORF7a crystal structures shows that the deletion is concentrated on the protein ectodomain structure (comprising β strands 3 to 7), the full transmembrane domain and the ER retention motif (Figure 1C). Due to the severe impact on the protein tridimensional structure, we suggest that the 58301_ORF7a_∆190 deletion potentially impacts severely the protein structure and impairs its function, since the loss of the carboxy-terminal transmembrane domain would prevent ORF7a interaction with host molecules (Figure 1C). In addition to that, we noticed 4 altered amino acid residues LLRQ, at positions 40 to 44 (Figure 1C). To access the coverage and read depth of SARS-CoV-2 sequenced genomes of all family members, we mapped the generated NGS reads to Wuhan RefSeq, as shown in Figure 1D. All five family members had exclusively the SARS-CoV-2 ORF7a_∆190 observed by the gap in the final third of the 3’ portion of the viral genome corresponding to the ORF7a region. No other deletion was observed on the genomes analyzed.

Nevertheless, data from the assembled virus genomes submitted to Nextclade on the Nextstrain website showed the same variants indicating that the viruses infecting all the family members belong to the B.1.1.33 lineage. Appendix A shows the single nucleotide variants common to all five samples when compared to the original Wuhan SARS-CoV-2. The variants of each sample and the position along the SARS-CoV-2 genome are represented. Changes from cytosine to thymine are found in ORF1ab and 5’UTR. Changes from C to T, A to G, T to C, G to A, G to C and T to C were observed in the structural and accessory proteins. Nine out of 12 of these single nucleotide variants generated change in the amino acids in ORF1a (2), ORF1b (1), S (2), ORF6 (1) and N (3). Therefore, the persisted ORF7a_∆190 virus was the same genome variant that infected the five subjects with mild COVID-19 symptoms, independent of the subject’s age.

### 3.2. Identification of SARS-CoV-2 ORF7a Deletion on Co-Workers

We continued to track new deletions and analyzed 100 samples collected by CTD, and we came across a group of co-workers (Appendix A) with larger deletions on the ORF7a region. PCR products encompassing the ORF7a region revealed different fragment sizes (Figure 2A). We observed that sample 16538 carried two fragments with similar proportions, suggesting that it contained at least two sets of viral genomes, one with a deletion on the ORF7a gene and another without. Sample 16991 revealed a slightly larger fragment, suggesting a smaller deletion than 16538. To investigate if the sample was contaminated with two human genomes, the human DNA was profiled using Short Tandem Repeats (STR) for human identification (Appendix A). The sample profile indicated to be from a single human donor, ruling out the contamination hypothesis, thus confirming the presence of two viral genomes. Serial dilutions of this clinical viral sample (16538_ORF7a_∆365) were used to inoculate the cell culture. After the cytopathic effect, the supernatant was collected to determine the sample’s prevalence of the SARS-CoV-2 genomes. Figure 2B indicates the ORF7a PCR products, corresponding to SARS-CoV-2 ORF7a_∆365 and the SARS-CoV-2 wild-type of the amplified fragments from the 10^–1^, 10^–2^, 10^–3^, and 10^–4^ serial dilution of the clinical sample. The two PCR fragments (around 500 bp and 900 bp) are present in 10^–1^ and 10^–2^, and the upper fragment alone is present in 10^–3^ and 10^–4^, suggesting that virus genomes with wild-type ORF7a are prevalent in the clinical sample (Figure 2B). Viral isolates were obtained by plaque purification of sample 16538 (Figure 2C). The SARS-CoV-2 ORF7a_∆365 (4 plaques—P3, P5, P6 and P7) and the counterpart ORF7a_wt SARS-CoV-2 (3 plaques—P2, P4 and P11) were distinctly isolated (Figure 2C). Sanger sequencing using amplified RT-PCR fragments of these samples confirms these deletions’ exact size (Figure 2D,E). Sample 16991_ORF7a_∆339 displays a 339-nt deletion in ORF7a (Figure 2D). Clinical samples and viruses isolated from cell culture (lanes P2 and P5, Figure 2B) of 16538 showed the 365-nt deletion (16538_ORF7a_∆365) and a non-deleted virus genome (the counterpart 16538_ORF7a_wt) (Figure 2E). These deletions may impair the complete function of the ORF7a protein.

High-coverage sequencing of these genomes showed that the three individuals (16305, 16991_ORF7a_∆339 and 16538_ORF7a_∆365) were infected with the same SARS-CoV-2 variant. Appendix A shows the nucleotide variants of the virus genome for these three samples. The common single nucleotide variants present in these samples indicate transmission of the same genome SARS-CoV-2 variant with the detected deletion in ORF7a.

### 3.3. ORF7a Deletions Impact on sgRNAs

To investigate if the ORF7a deletions (ORF7a_∆190, ORF7a_∆339, ORF7a_∆365) would impact the SARS-CoV-2 sgRNA population, we designed primers that would encompass the leader sequence fused to the correspondent ORF of each accessory protein that further enables the identification of sgRNAs (Figure 3A). To specifically confirm the ORF7a deletions, we designed one primer (27,511-27,491) upstream of the 190-nt deletion region but within 339-nt deletion and one primer (27,531-27,512) in both deletion regions (Appendix A). PCR amplification from different samples can be visualized in Appendix A. As expected, we did not detect any PCR amplification when the ORF7a contained the deletion covering the primer binding sites. Bands of amplified products of the sgRNAs of ORF7a, ORF8 and N (indicated in Appendix A) were further isolated, sequenced and mapped to a sgRNA template containing the leader sequence of 75-nt, and the coding sequences of each ORF (Figure 3B). Sequences confirmed that downstream of ORF7a did not change its size displaying the same sequence among the different samples (wt, ∆190 and ∆339). In contrast, ORF7a-sgRNAs carried out the expected deletions from the corresponding samples. Note that some of these fragments revealed to be non-canonical (or aberrant) sgRNAs.

### 3.4. SARS-CoV-2 Genomic Data

To compare the deleted SARS-CoV-2 sequences with other genomes available in public databank from the same period, we analyzed 208 published genome sequences deposited on GISAID from April to September of 2020 for the State of Rio de Janeiro and from Uruguay, and all samples belong to B.1.1.33 lineage. A phylogenetic analysis was performed, and the color code of the resulting tree is presented (Appendix A). All three groups of deleted ORF7a were clustered separately meaning all three clusters are intra-geographic correlated by place of home or work.

### 3.5. Replication Capacity of SARS-CoV-2 with ORF7a Deletions

Vero E6 cells were infected with SARS-CoV-2 ORF7a_Δ190 (58301_ORF7a_∆190) and a SARS-CoV-2_ORF7a_wt (11785_wt) isolated from samples from the same familiar cluster, and representing wildtype (11785), and ORF7a_Δ190 (58301) pure virus stocks. After 24, 48 and 72 hpi, cells supernatant was collected for virus titration and RT-qPCR. SARS-CoV-2 ORF7a_Δ190 (58301_ORF7a_∆190) led to extensive cytopathic effect similar to SARS-CoV-2_wt (11785) after 48 hpi (Figure 4A). No difference between SARS-CoV-2_ORF7a_Δ190 and SARS-CoV-2_wt progeny infectious virus production at 24 hpi was observed (Figure 4B) although higher levels of gRNA were observed (Figure 4C), which gave an RNA copy number/PFU ratio of 2.8 for the former and 1.1 for the later. SARS-CoV-2 ORF7a_Δ190 showed lower progeny infectious virus than SARS-CoV-2_wt at 48 and 72 hpi (Figure 4B) while increasing the RNA copy number/PFU ratio to 82.5 and 29.6, respectively. This ratio for the SARS-CoV-2_wt was 4.6 and 6.4, respectively (Figure 4B,C). These results suggest that SARS-CoV-2 ORF7a_Δ190 produced more virus particles than SARS-CoV-2_wt at 72 hpi when measured by the presence of genomic viral RNA in the cell-free supernatant. (Figure 4C). Probably these viruses are defective since SARS-CoV-2 ORF7a_Δ190 progeny infectious virus was smaller than SARS-CoV-2_wt at 72 hpi (Figure 4B). This result shows that SARS-CoV-2 ORF7a_Δ190 can replicate in SARS-CoV-2 susceptible cell cultures similarly to wild-type virus, a less infectious virus progeny is produced during virus spread. We also attempted to compare the replication kinetics of (ORF7a_∆365 and the counterpart ORF7a_wt) from sample 16538_ORF7a_∆365 (Figure 2C). Although we were able to plaque purify these viruses, we failed to grow virus stocks from isolated plaques in different cell cultures (Appendix A).

## 4. Discussion

We report the following deletions of ORF7a in the genome of SARS-CoV-2: ORF7a_∆190, ORF7a_∆339 and ORF7a_∆365. The ORF7a_∆190 genome was found among five individuals belonging to the same family and ORF7a_∆339, and ORF7a_∆365, were found among a group of co-workers. Variant analysis suggests that the same variant was infecting each of the two groups in both cases. These deletions were observed in the early days—April to September 2020 of the pandemic in Rio de Janeiro State. According to the tree clusters, these samples correlate to individuals by place of home or work. Recent data show the spread of SARS-CoV-2 infection in the whole country during pandemics [39,40,41,42,43,44].

So far, deletions in ORF7a have been found in samples in the USA [26,45], Thailand [27] and Uruguay [25]. In addition, a few cases have reported the complete loss of ORF7a [46,47]. These deleted variants belong to different strains, such as B.1; B.1.1140, B.1.1.129, B.1.1234 and B.1.1.33, to which our samples and the ones from Uruguay belong. The amino acid content found by alignment suggests that the protein’s carboxy terminal is truncated.

The family members with ORF7a_∆190 deletions were all infected with the same virus based on SNP variants analysis (Appendix A), indicating that the virus can still infect and replicate with this significant deletion. The 190-nt deletion causes a premature truncation of the protein’s ectodomains, the loss of the transmembrane and the cytoplasmic domain, and the alteration of four amino acid residues. Truncations in the ORF7a C-terminus have been shown to occur frequently amongst different lineages, and it limits the suppression of the type I IFN signaling in vitro [48]. The ORF7a_∆190 virus replicated in cultures of SARS-CoV-2 susceptible cells, although with lower infectious viral titers compared to SARS-CoV-2_wt, suggesting the truncation of the carboxy-terminal may impact the ability of the virus to replicate in vitro. Interestingly, SARS-CoV-2 ORF7a_Δ190 produced more non-infectious virus particles than SARS-CoV-2_wt (Figure 4C), suggesting that ORF7a has a role in SARS-CoV-2 infectivity. It has been shown in previous studies that deletions on the transmembrane domain of the ORF7a gene result in growth defects in vitro [48], and small deletions (six and seven nucleotides long) in this domain were able to delay replication of SARS-CoV-2 [30]. A different study showed that reporter-expressing recombinant SARS-CoV-2, which expresses all SARS-CoV-2 genes, had higher reporter gene expression levels than recombinant viruses in which the reporter gene replaced the ORF7a gene [49].

In contrast, deletions on ORF7b/8 encompassing an area of ORF7b and ORF8 showed significantly higher replicative fitness in Vero-E6 cells than in the wild type. At the same time, no difference was observed in patient viral load, indicating that the deletion variant viruses retained their replicative fitness [50]. The ORF7a deletion effect on replication seems to vary according to the virus lineage and can generate more viral progeny [30].

We observed that the 339-nt deletion removes almost all the ORF7a gene, and the 365-nt deletion eliminates the coding sequencing of ORF7a. SNV analysis (Appendix A) suggested that the same virus infected the group of co-workers (16305_wt, 16991 (ORF7a_∆339) and 16538 (ORF7a_∆365 and the wt counterpart). From our analysis, we cannot conclude whether the ORF7a_∆339 virus infected the contacts producing an even larger deletion of ORF7a_∆365, or if both viruses lost the ORF7a gene or part of it after infection spread. In both samples, the ORF7a deletions are visible through PCR (Figure 2A), meaning that the virus has at least two sets of viral genomes, one with a deletion on ORF7a and the other without deletion. However, after diluting the sample ORF7a_∆365 more than a thousand times, only its counterpart, ORF7a_wt genome, was isolated in Vero E6 cells (Figure 2C). It suggests that the wild type of virus genome was prevalent in the clinical sample.

Cell cultures were inoculated with SARS-CoV-2 obtained from the clinical sample 16538_ORF7a_∆365 containing both fragments (ORF7a_∆365 and its counterpart ORF7a_wt), and viruses were isolated (Figure 2C). However, the production of infectious particles was not high enough to proceed with the virus profile (Appendix A). We speculate that both genomes (ORF7a deleted and the counterpart ORF7a_wt) are necessary to maintain virus spread in culture cells, which could involve recombination mechanisms. No additional mutation in this ORF7a_wt isolate was observed when compared to other B.1.33 viral isolates that could explain these results. Further studies are necessary to understand better the impact of complete ORF7a deletion on virus replication and transmission.

The frameshift of 190-nt in ORF7a deletion (SARS-CoV-2 ORF7a_Δ190) causes a premature truncation of the protein ectodomain losing the transmembrane and the cytoplasmic protein domains as described for this transmembrane protein [51]. It was found that the ORF7a ectodomain structure interacts with high efficiency with CD14+ monocytes in human peripheral blood, compared to pathogenic protein SARS-CoV ORF7a, and SARS-CoV-2 ORF7a coincubation with CD14+ monocytes upregulated the production of proinflammatory cytokines, including IL-6, IL-1β, IL-8 and TNF-α [24]. A large part of the ectodomain structure is lost with the deletions, especially the one found in our sample 58301_ORF7a_∆190 (Figure 1C), which could impair the proper functioning of ORF7a. It is known that monocytes are vital contributors to cytokine storms in COVID-19 [52] and may suggest that a malfunctioning ORF7a could be related to mild COVID-19.

Some reports gathered evidence that ORF7a sgRNAs are expressed in the double membrane complex [12]. Authors report detecting sgRNAs containing the leader sequence in swab samples from 81 individuals and that ORF7a followed by N and E the most abundant sgRNAs. The authors speculate that although not fully understood, it is believed that both replication and transcription occur in the so-called double-membrane vesicles in the cytoplasm of infected cells and that detection of SARS-CoV-2 sgRNAs in diagnostic samples up to 17 days after initial detection of the infection, which evidences its nuclease resistance and cell membrane protection, suggesting that the detection of sgRNAs in such samples may not be a good indicator of replication/infection of active coronavirus. In another report, SARS-CoV-2 sgRNA is detected in Vero cells infected at varying times of infection. The RPKM values normalized according to the N gene, which is expressed significantly, were followed by the other two accessory proteins, ORF8 and ORF7a [53]. Our work successfully identified sgRNAs through RT-PCR followed by Sanger sequencing. The PCR fragments associated with N, ORF7a and ORF8 were the most abundant in concordance with other reports [53,54].

Nevertheless, we noticed a few aberrant or non-canonical templates in our samples. These TRSB-independent or even a non-TRS-dependent events are described for SARS-Cov-2 [55,56,57]. The global dynamics of the coronavirus subgenomes and their regulatory pattern are still unclear [57,58]. Therefore, the profiling of sgRNA in deleted ORF7a genomes should be further investigated.

High similarity between the ORF7a of the SARS-CoV (2002) and SARS-CoV-2 is found [59], showing that ORF7a is a viral antagonist of tetherin, also known as BST-2 and CD-317 [60]. Other viruses, such as HIV and Ebola, have proteins that block or antagonize the role of tetherin, which restricts the budding of the virus from the cell [61,62,63]. It was shown that cells infected with SARS-CoV with ORF7ab deletion display increased sensitivity to BST-2 compared to the control infected with SARS-CoV in which ORF7ab was present. It was also found that cells overexpressing BST-2 infected with SARS-CoV and SARS-CoV without ORF7ab gene, decrease in replication and leads to significant restriction, confirming the role of ORF7a as a BST-2 inhibitor acting as a blockage of BST-2 glycosylation [60] and also inducing apoptosis by interacting with BCl-X [64,65]. Therefore, SARS-CoV-2 ORF7a may play a significant role in the virus–host interaction, and studies using SARS-CoV-2 deleted ORF7a may add new information to understand replication mechanisms better.

## 5. Conclusions

We identified three different sizes of ORF7a deletion in upper respiratory clinical samples from individuals infected with SARS-CoV-2 circulating in family relatives or co-workers’ contacts. In silico analysis suggests that the deletions impair protein proper function and in vitro analysis revealed that partial deletions on the ORF7a gene (SARS-CoV-2 ORF7a_Δ190) can replicate in culture cells similarly to wild-type viruses at 24 hpi, but it is less infectious after 48 hpi. Although the ORF7a_∆339 virus was capable of infecting the contacts producing an even larger deletion of ORF7a_∆365 (or both viruses lost the ORF7a gene or part of it after the infection spread), the in vitro production of infectious particles was not enough to proceed with the analysis of the virus profile in cell cultures. These findings contribute to understanding SARS-CoV-2 phenotypes, such as replication and evolutionary fitness, and insights into the role of SARS-CoV-2 ORF7a in virus–host interactions.

## Figures and Tables

**Figure 1 viruses-15-00801-f001:**
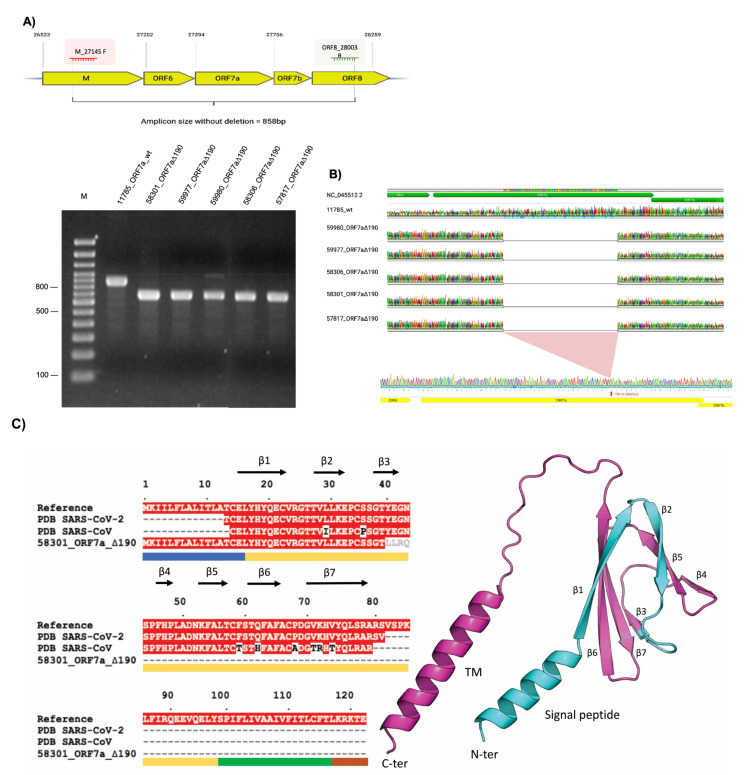
A 190-nucleotide deletion in the ORF7a gene (ORF7a∆190) and impact on the protein structure. (**A**) Top: Amplification and confirmation of 190-nt deletions amongst family members using primers forward (M_27145 F) and reverse (ORF8_28003R) (see scheme) indicating the positions on the SARS-CoV-2 genome. Bottom: ORF7a 190-nt deletions visualization in five family members on agarose gel (2%) stained with ethidium bromide revealed that the PCR products target the ORF7a gene. 100bp Molecular Weight Marker (lane M), 11785_ORF7a_wt, 58301_ORF7a_∆190, 59977_ORF7a_∆190, 59980_ORF7a_∆190, 58306_ORF7a∆190, 57817_ORF7a∆190. (**B**) Electropherograms from Sanger sequencing of the ORF7a PCR products from samples with 190-nt deletions: 59980, 59977, 58306, 58301 and 57817, the wt (without 190-nt deletion) 11785 aligned to the SARS-CoV-2 reference sequence (NCBI RefSeq SARS-CoV-2 genome sequence, NC_045512.2). (**C**) ORF7a deletions and impact on the protein structure. Left: Multiple sequence alignment of the reported ORF7a deletion (58301_ORF7a_∆190), and the corresponding sequences for SARS-CoV-2 and SARS-CoV ectodomain structures (PDB IDs: 7ci3 and 1xak, respectively) in respect of the reference sequence (NCBI RefSeq for SARS-CoV-2 ORF7a protein, YP_009724395.1). Identical residues are shaded in red; mutations from SARS-CoV to SARS-CoV-2 are colored in black and ORF7a altered residues are colored in grey. The labeled arrows indicate the secondary structure observed on the crystal structures. The different protein domains are indicated by the color bars below the alignment: N-terminal signaling region (residues 1–15, blue), Ig-like ectodomain (residues 16–96, yellow), hydrophobic transmembrane domain (residues 97–116, green) and ER retention motif (residues 117–121, brown). Right: Ribbon representation of the tridimensional SARS-CoV-2 ORF7a structure. The full-length structure of the ectodomain is colored in cyan. The β-strands are assigned numerically from the N-terminus to the C-terminus, corresponding to the alignment above. The deletion presented by the 58301_ORF7a_∆190 sample is colored in magenta, in both structures, covering a significant portion of the protein ectodomain, comprising the β sheets 3 to 7, the full transmembrane and ER retention motif. (**D**) Sars-CoV-2 whole genome sequencing of samples belonging to the five family members mapped to the reference sequence, aligned and visualized on IGV viewer. ORF7a region displays gaps evidencing the 190-nt deletion in all five samples.

**Figure 2 viruses-15-00801-f002:**
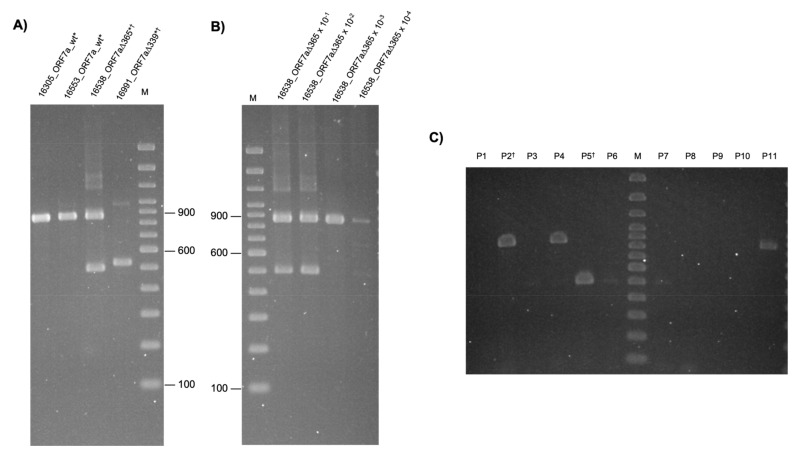
Amplification and confirmation of 339 and 365-nt deletion amongst co-workers. (**A**) ORF7a 339-nt and 365-nt deletions visualization in a group of co-workers on agarose gel (2%) stained with ethidium bromide revealed the PCR products targeting ORF7a gene: 16305_ORF7a_wt, 16553_ORF7a_wt, 16538_ORF7a_∆365, 16991_ORF7a_∆339; 100bp Molecular Weight Marker (lane M). (**B**) Prevalence of viral genomic deletion analyzed through ORF7a PCR products from serial dilutions (10^–1^–10^−4^, lane 2–5, respectively) of sample 16538_ORF7a_∆365 viral stock and visualized on agarose gel (2%) stained with ethidium bromide. The two PCR fragments are present in 10^−1^ and 10^−2^, and only the non-deleted fragment is present in 10^−3^ and 10^−4^. (**C**) Eleven virus plaques were isolated from sample 16538. RNA extraction, reverse transcription, and PCR using primers that target the ORF7a region were performed. PCR products were visualized on agarose gel (2%) stained with ethidium bromide. Each lane corresponds to a virus plaque (P1–P11) derived from sample 16538. Virus plaques P2, P4 and P11 correspond to the virus without deletion in the ORF7a region, and virus plaques P5, P6 and P7 correspond to the virus with 365-nt deletions in the ORF7a region (ORF7a_∆365). M = 100 bp Molecular Weight Marker. (**D**) Electropherograms from Sanger sequencing of the ORF7a PCR products from sample 16991_ORF7a_∆339 aligned to SARS-CoV-2 reference and its deletion-resulting sequence (**E**) Electropherograms of both ORF7a PCR products isolated on a gel, from clinical samples and viruses isolated from cell culture (lanes P2 and P5, (**B**)) of 16538 ORF7a WT and the ORF7a_∆ 365 of 16538_ORF7a_∆365 and the non-deleted counterpart of 16538 * Samples submitted to NGS sequencing † Samples submitted to Sanger sequencing.

**Figure 3 viruses-15-00801-f003:**
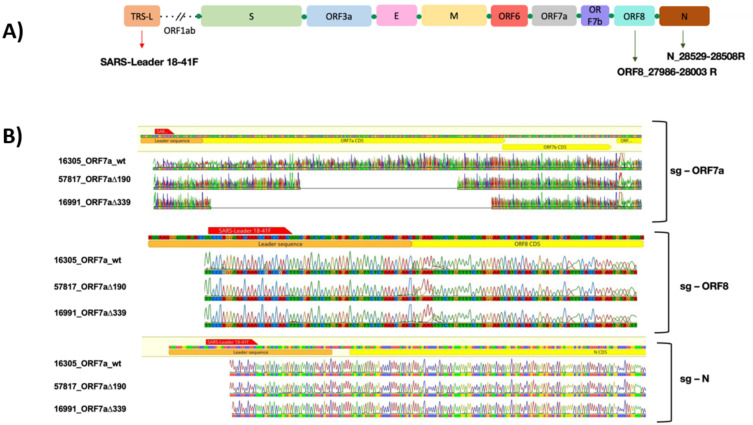
Amplification and examination of sgRNAs from samples with and without deletions on ORF7a. (**A**) Schematic illustration of SARS-CoV-2 genome encompassing TRS-L leader sequence (light orange box) and the primers’ position for sgRNAs amplification. Green dots represent the TRS-B sequences at the 5′ of each ORF of the structural and accessory proteins. (**B**) Sanger sequencing visualization of sgRNAs fragments from samples ORF7a_wt, ORF7a_∆190 and ORF7a_∆339 as illustrated in Appendix A. The light orange box indicates the 75-nt leader sequence, the red bar represents the forward primer SARS-Leader 18-41F, yellow boxes represent coding sequences (ORF7a, ORF8 and N), the reverse primers are not shown.

**Figure 4 viruses-15-00801-f004:**
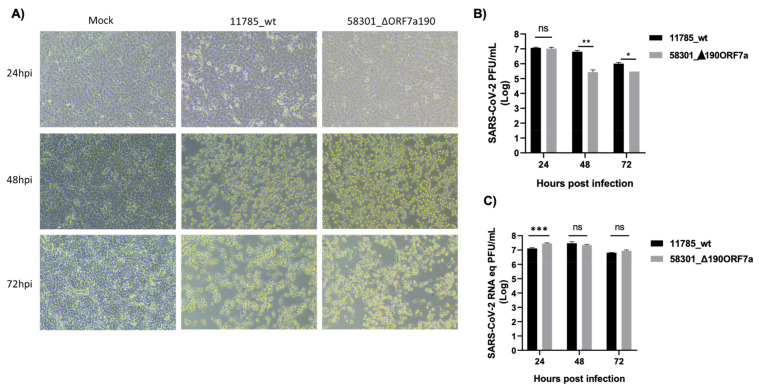
SARS-CoV-2 ORF7a 190-nt deletion reduces virus infectivity in Vero E6 cells. (**A**) Vero E6 cells were infected at MOI of 0.1 of SARS-CoV-2_wt (11785_wt) or SARS-CoV-2_ORF7a_∆190 (58301_ORF7a_∆190). (**B**) Cell supernatant was collected at 24, 48 and 72 hpi and used for virus titration in Vero E6 cells. Virus titration (mean ± SD) was plotted in a graphic. The virus titles at each time point were compared using multiple unpaired t-tests. PFU = plaque forming units (**C**) Cell supernatant was collected at 24, 48 and 72 hpi and used to perform RT-qPCR for SARS-CoV-2 N. SARS-CoV-2 RNA equivalent to PFU (mean ± SD) was plotted in the graphic. The virus titrations at each time point were compared using multiple unpaired *t*-tests. Results are from three independent experiments. * *p* < 0.05; ** *p* < 0.01; *** *p* < 0.001; ns indicates no significant difference (*p* > 0.05).

## Data Availability

All DNA sequences generated during the current study are available in the GISAID EpiCoV repository (https://www.gisaid.org/, accessed on 14 January 2023) and in the NCBI database (https://www.ncbi.nlm.nih.gov/nuccore/, accessed on 14 January 2023) under the following accession IDs: 16538 ORF7a_∆365 and its counterpart ORF7a_wt (EPI_ISL_15977591; OQ430693), 16553_ORF7awt (EPI_ISL_16093680), 16305_wt (EPI_ISL_15977590; OQ430692), 16991_ORF7a∆339 (EPI_ISL_15977592; OQ430694), 58301_ORF7a∆190 (EPI_ISL_16093682; OQ430699), 59977_ORF7a∆190 (EPI_ISL_16093683; OQ430700), 59980_ORF7a∆190 (EPI_ISL_16093684; OQ430701), 58306_ORF7a∆190 (EPI_ISL_16093685; OQ430702), 57817_ORF7a∆190 (EPI_ISL_16093686; OQ430703).

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
