# Peer review of "Evaluation of SARS-CoV-2 ORF7a Deletions from COVID-19-Positive Individuals and Its Impact on Virus Spread in Cell Culture"

_viruses, 2023, doi:10.3390/v15030801_

Round 1
Reviewer 1 Report
Title: Evaluation of SARS-CoV-2 ORF7a deletions from COVID-19-positive individuals and its impact on virus spread in cell culture
Summary: This study investigated the effects of genomic deletions in the ORF7a coding region of SARS-CoV-2 isolated from human clinical samples on cell culture. However, it appeared that the authors failed to provide evidence enough to support the conclusions drawn in the manuscript.
First, the experimental methods are included throughout the results section. The manuscript should be rewritten. Especially, In the lines 262-265, page 6, the authors claimed that ‘…, the full 262 transmembrane domain and the ER retention motif (Figure 1C). The 58301_ORF7a_∆190 263 deletion impacts severely the protein structure and impairs its function, since the loss of 264 the carboxy-terminal transmembrane domain will impact Orf7a interaction with host 265 molecules (Figure 1C).’ However, no results of protein function analysis and host interactions were provided in the manuscript.
Second, in Figure 4B and C, how can you explain the different patterns of the detected amounts of infectious particles and RNA genome levels in cells? Probably, isn’t it the reason that the higher PFUs of deletion mutant at 48 hpi than wild-type due to relatively high MOI (0.1)? Hence, did no cells remain for progeny virions to infect after multiple rounds of viral infection?

Author Response
We thank the editor and the reviewers for their time and positive and constructive feedback. We provided a document stating point-by-point responses to the comments, and highlights in yellow indicate the modifications in the manuscript. Please see the attachment.

Reviewer 2 Report
suffers -> undergoes or "switches template to" (line 60)
I noticed missing citations for IGV (line 207) 10.1038/nbt.1754 and nextstrain (line 209) https://doi.org/10.1093/bioinformatics/bty407 and IQ-TREE2 line 210) https://academic.oup.com/mbe/article/37/5/1530/5721363
Please look for any other missing citations of tools used. In future work I suggest creating an executable workflow
Figure 1C - direct labeling of what is deleted (pink) what is alphafold and what is crystal structure would be helpful.
Figure 1 D - this view does not display spanning reads. While it's not necessary given your sanger data from figure 1B, you could a different aligner to correctly "splice" across this junction and identify the precise boundaries. In similar samples I was only able to identify the precise boundary of similarly large ORF7 deletions precisely using bbmap. BWA-mem, minimap, bowtie2 or hisat 2 failed to map or soft-clipped deletion-spanning reads. Also, consider loading a GFF track to your IGV image to show the genomic features.
line 325: Do the two sequence with differnet ORF7 deletions have the same constellation of mutations?
Figure 2 B (line 349) seems to be missing some tracks (i see only 16991 and 16538 in my copy). Seems odd that the figure proceed in order A -> C -> B instead of A B C.
line 364: the text states that these 4 variants are the "same lineage", However, Supp table S4 shows 16553 with 6 mutations not shared by the others. How can this be reconciled?
line 386:
Searching gisaid for such deletions is not practical for people who do not have gisaid accounts. To enable meaningful reproducibility, could you please upload the reads (bam or fastq format) in addition to the consensus sequences to a public INSDC database (NCBI, EBI,...)?
Looking at your sequences... I do not see a deletion in 16538. Perhaps the GISAID entry corresponds to the full length sequence and the sequence with the deletion is not present or not listed?
Author Response

(The authors gave the same response as above.)

Reviewer 3 Report
The authors in this manuscript identify three orf7 deletion virus from people contracted with COVID-19 and did further replication characterization of those deletion virus in comparison to WT. The authors concluded that virus bearing orf7a deletion (orf7a_del190, potentially the del365) had reduced infectivity. I think this study is interesting, but more characterization need to be done.
1. Figure 3 is confusing. From Fig.3, the authors concluded in the results that sgRNA downstream of orf7a did not change size, but fragments associated with sgRNA of genes upstream orf7a showed a decrease. Have the authors cut those band and send for sequencing to confirm whether those bands are the expected viral products? Compared to WT, there was an extra band lower than potential orf7a band. However, how do the authors know the lower band was from the sgRNA of genes upstream orf7a? Although the authors try to use the primer set specifically target sgRNAs, this primer pair could also detect large deletion not within sgRNAs. To make this conclusion more convincing, the authors need to cut the bands and send for sanger’s sequencing to confirm the identity of viral products. In addition, the authors need to add mocks in this figure.
2. For figure 4, did the authors use purified orf7a deletion virus or the virus amplified from the clinical samples? As the viruses from the clinical sample contained both orf7a and WT with unknown ratios. If the authors did use the unpurified virus to do the infection under same MOI, the experiments need to be repeated using the purified virus containing either only WT or only orf7a deletion.
3. There is an inconsistence between Fig. 4B and Fig. 4C at 72hpi. The authors claimed that this is because “these viruses (I assume del190 viruses are defective). However, there was no differences between WT and del190 at 24hpi, suggesting in the beginning those two viruses have similar infectivity. Did the author treat supernatants with RNases before performing RT-PCR? Is it possible that the higher RNAs were leaking from cells rather than the genomes packaged in the infectious virions?
Author Response

(The authors gave the same response as above.)

Round 2
Reviewer 1 Report
No further issues.
Author Response
We appreciate your efforts made to improve our manuscript.
Thank you
Reviewer 3 Report
The authors did not answer my first question in regard to Figure 3 gel. Authors only sequenced the PCR products targeting orf7 or orf7a region in fig.2. However, the fig.3 aims to test whether identified deletions impact subgenomic RNAs and that's why authors used a primer pair containing one in TRS-L and one in N. This is different from the PCR from fig.2 and thus the deletion should be further confirmed in those bands by sanger's sequencing again. Can authors run regular gel and cut targeted bands and sequence the expected deletion region? It is important to know whether the deletions are only contained in orf7 (orf7a) sgmRNAs or sgmRNAs before orf7? If it is the latter, it will further prove the deletions are in the viral genomes. If authors cannot do this experiment, at least they should do western blot to show the deletion or defect expression of orf7a on the protein level.
Author Response
We appreciate your efforts made to improve our manuscript.
We carefully studied your recommendations and made point-by-point changes to the article as a result.

Round 3
Reviewer 3 Report
The authors answered my question. Couple minor suggestions regarding the new added data and writing. Please revise.
1. In Fig. S2B, the size for your ladder (M) need to be indicated so we would know the size of bands.
2. I assume the sequences in Fig.3B are from certain bands shown in Fig. S2B. Please use different symbols (or numbers) to indicate which band in Fig. S2B corresponds to which sequence in Fig. 3B. And please also indicate this in the figure legend.
3. I suggest to revise the paragraph in LN388-397 as follows:
we designed primers that would encompass the leader sequence fused to the correspondent ORF of each accessory proteins that further enables the identification of sgRNAs (Figure 3A). To specifically confirm the ORF7a deletions, we designed one primer (27511-27491) upstream of the 190nt deletion region but within 339nt deletion and one primer (27531- 27512) in both deletion regions (Figure S2A). PCR amplification from different samples can be visualized in Fig S2B. As expected, we did not detect any PCR amplification when the ORF7a contained the deletion covering the primer binding sites. Bands amplifying the sgRNAs of ORF7a, ORF8, and N using ? primers (indicated in Figure S2B) were further isolated, sequenced, and mapped to a sgRNA template containing the leader sequence of 75 nt and the coding sequences of each ORF (Figure 3B). Sequences confirmed that downstream of ORF7a did not change its size displaying the same sequence among the different samples (wt, ∆190, and ∆339). In contrast, ORF7a-sgRNAs carried out the expected deletions from the corresponding samples. Note that some of these fragments revealed to be non-canonical (or aberrant) sgRNAs (data not shown).
Author Response
Thank you for your comments.
